# The Facile Solid-Phase Synthesis of Thiazolo-Pyrimidinone Derivatives

**DOI:** 10.3390/molecules30020430

**Published:** 2025-01-20

**Authors:** Shuanghui Hua, Jimin Moon, Taeho Lee

**Affiliations:** Research Institute of Pharmaceutical Sciences, College of Pharmacy, Kyungpook National University, 80 Daehak-ro, Buk-gu, Daegu 702-701, Republic of Korea; ace_goh@naver.com (J.M.);

**Keywords:** thiazole, solid-phase synthesis, chemical library

## Abstract

A thiazolo-pyrimidinone derivative library was developed through a facile solid-phase synthesis method. For the reaction, the thiazolo[4,5-*d*]pyrimidin-7(6*H*)-one structure was synthesized through efficient Thorpe–Ziegler and cyclization reactions. The thiazolo[4,5-*d*]pyrimidin-7(6*H*)-one derivative library with a diversity of three had a total of four synthesis steps and 57 compounds. In addition, the yield per synthesis step was 65–97%, which was very high. The developed synthesis method and compounds will be used to find compounds with biological activity through the thiazole derivative structure–activity relationship.

## 1. Introduction

The importance of compound diversity in drug development has been increasingly emphasized in the search for hit compounds, particularly as DNA-encoded libraries (DELs) and high-throughput screening (HTS) technologies continue to advance. In this context, modifying pharmacologically validated structures has become one of the key strategies in drug discovery [1,2,3,4]. Among these, heterocyclic scaffolds play a crucial role as target compounds for a variety of diseases [5,6,7,8,9,10]. Many pentagonal heterocyclic compounds are being used as key structures to identify hit compounds, including thiazole derivatives with α-amylase inhibitory activity [11], thiophene derivatives with potential anticancer efficacy [12], imidazole derivatives with potential antimalarial activity based on SAR studies [13], and oxazole derivatives with potential hypoglycemic effects [14]. Accordingly, our research focuses on constructing a diverse range of compounds by synthesizing heterocyclic compounds and building a compound library. However, one of the challenges when working with heterocyclic structures is the difficulty of purification, which often arises due to poor physical properties during synthesis. This issue is particularly evident with five-membered heterocycles, such as thiazole, imidazole, and thiophene, where additional difficulties in terms of synthesis may occur. To address this challenge, we have utilized solid-phase synthesis, which allows for the rapid and efficient production of large numbers of compounds while bypassing the traditional purification process through solvent-based washing [3a]. Until now, various synthesis methods, such as heterocyclic compounds and peptides using solid-phase synthesis, have been introduced. Solid-phase synthesis methods are used in various synthesis fields, such as macrocycle synthesis [15], peptidomimetics synthesis with chirality [16,17,18,19], and pyrimidine derivative synthesis using microwaves [20], and are worth utilizing. Despite these advantages, solid-phase synthesis presents several limitations due to the restrictive conditions imposed by factors such as the base, solvent, reagent, and temperature. As such, optimizing the reaction conditions is critical to the successful synthesis of the desired compounds. For the above reasons, we are conducting research to select structures that are expected to have pharmacological activity to build a heterocyclic chemical library and then build a chemical library through solid-phase synthesis through the optimization process of solution-phase synthesis. In particular, we are interested in building a library of thiazole derivatives among many five-membered heterocyclic compounds, and many studies related to the pharmacological activity of thiazole have been conducted. Recently, thiazolo-pyrimidinone derivative **2** has emerged as a promising drug candidate for improving cognitive impairment, with structure–activity relationship (SAR) studies identifying it as a key compound in rodent in vivo models [21]. Additionally, thiazolo-aminoamide derivatives **3** have shown potential as inhibitors of protein interactions with NIMA1 (Pin1) [22]. Furthermore, thiazolo-pyrimidine derivatives **4** have demonstrated antiproliferative activity in human cancer cell lines, with specific efficacy observed in the MGC-803 gastric cancer cell line, with IC_50_ values of 5.13 and 4.64 μM (Figure 1) [23]. Over the years, a variety of bioactivities have been associated with thiazole derivatives, highlighting their potential as effective core scaffolds for drug development. Along with previous studies, we are conducting research to build a chemical library to find a new hit compound. In our previous work, we have successfully constructed libraries of five-membered heterocyclic derivatives, including 4-substitued xanthine derivatives, 3-substituted thiazole derivatives, and a 3-substituted thiophene derivative, using solid-phase synthesis (Figure 2) [24,25,26]. As discussed previously, based on several structure–activity relationship (SAR) studies of thiazolo-aminoamide **3** and the bioactivity of the thiazole-fused heterocyclic compounds **2**, **4**, we propose a method for synthesizing a thiazolo-pyrimidinone derivative library that has not yet been synthesized in solid-phase synthesis.

## 2. Results and Discussion

The process of the overall solution-phase synthesis method is summarized in Figure 1. To begin, potassium cyanocarbonimidodithioate **5** and chloroacetamide **6a**, a diversity element, were reacted to synthesize intermediate **7-int I**, after which the effective Thorpe–Ziegler reaction conditions were optimized. After the Thorpe–Ziegler reaction through the base, the thiol of **7-int II** was methylated using methyl iodide to synthesize compound **7a.** Moreover, the compound **7a** was synthesized through a three-step one-pot reaction without isolating intermediates **7-Int I** and **7-Int II**. First, hydroxide bases such as NaOH, KOH, and LiOH were tested. When 1 M NaOH was used as the base, the desired product **7a** was obtained with a yield of 57%. Using potassium hydroxide resulted in a lower yield of 41%, while lithium hydroxide produced the desired product with a yield of 98%. Additionally, when the solvent was changed to ethanol (EtOH) and sodium hydroxide was used as the base, the yield was significantly reduced to 16%. Other bases, including NaH, K_2_CO_3_, triethylamine (Et_3_N), and DBU, were also tested, but no effective conditions were found (Table 1).

Next, we optimized the solution-phase synthesis conditions for an efficient solid-phase synthesis method. Through various cyclization reaction conditions, the process of introducing diversity factors to the R^2^ position was optimized. Initially, the cyclization reaction was attempted under acidic conditions using triethyl orthoformate to synthesize the thiazolo-pyrimidinone core structure **8aa** by introducing the hydride of R^2^ (Table 2). The reaction was carried out using acetic anhydride as the solvent, without an acid catalyst, but the reaction did not proceed (Table 2, Entry 1). When *p*-toluenesulfonic acid (*p*-TsOH) was used as a catalyst, we evaluated the yield based on the solvent, with the best yield obtained using ethanol (EtOH) (Table 2, Entries 2–6). Next, when the acid catalyst was changed, the reaction with AlCl₃ was inefficient, whereas using BF₃ led to a yield of 64%, comparable to the yield obtained with *p*-TsOH. However, when camphorsulfonic acid (CSA) was employed, the best yield of 79% was achieved, and the methyl group (Me) at R^2^ also resulted in a satisfactory yield of 75% (Table 2, Entries 7–10). Additionally, to introduce an aromatic group at R₂, the thiazolo-pyrimidinone core structure **8aa** was successfully synthesized with a high yield of 80% through an oxidative cyclization reaction using iodine (Table 2, Entry 11). To substitute various building blocks at the R^3^ position, methylsulfide was oxidized using *m*-CPBA to form the sulfone, and butylamine was then substituted to synthesize the final compound **1aaa** with a 60% yield (Figure 1).

Next, solid-phase synthesis was carried out based on the optimized solution-phase conditions. Merrifield resin and the synthesized intermediate **7-Int II** were stirred in acetone to form thiazolo-aminoamide resin **12a**. After the reaction, the FT-IR spectrum showed a peak at 34,823,351 cm^−1^ corresponding to the amine (NH_2_) stretch and a peak at 1640 cm^−1^ corresponding to the amide (R-CO-NHR^1^) stretch. The reaction was also performed under parallel conditions in solution-phase synthesis to prepare thiazolo-pyrimidinone resin **13aa**. However, the reaction did not proceed as expected, and only a very small amount of the final compound **1aaa** was obtained. This issue was attributed to the poor swelling of the resin in polar solvents, such as ethanol (EtOH). To address this, we improved the swelling effect by mixing DMF and EtOH as co-solvents, which resulted in the disappearance of the 3351 cm^−1^ amine stretch in the FT-IR spectrum. In addition, when DMF was reacted with a single solvent, it required a high temperature of 100 °C and a long reaction time. However, when using a mixed solvent of DMF and EtOH, the reaction could proceed in a shorter time (Figure 3). We oxidized sulfide using *m*CPBA to substitute nucleophile at the R^3^ position of thiazolo-pyrimidinone resin **13aa**. In this process, new 1151 cm^−1^ and 1336 cm^−1^ sulfone stretches could be observed, and then **1aaa** could be synthesized in an 81% yield through the nucleophilic substitution reaction using butylamine as a nucleophile and triethylamine as a base. In addition, the thiol types of nucleophiles of R^3^ could be synthesized at a yield of 77%. To introduce the methyl group of R^2^, **13ab** was synthesized using triethylorthoacetate, and then butylamine was substituted with nucleophiles after oxidation of sulfide to synthesize **1aba** with an 85% yield (Figure 2). This reaction represents an optimized condition, with a 95% yield per step over a total of four steps, and we successfully constructed a derivative library by introducing various building blocks. For the diversity elements, an aromatic ring with functional groups at R^1^ and hydrogen and methyl groups at R^2^ was selected. Additionally, 10 nucleophiles, including amine and thiol groups at R^3^, were chosen to construct a thiazolo-pyrimidinone derivative library (Figure 4). The yields for the library are summarized in Table 3. Initially, when R^1^ was phenyl, high yields were obtained for all the nucleophiles, except diethylamine, regardless of the functional group at R^2^ (Table 3, Entries 1–19). Similarly, when R^1^ was *p*-anisidine, the yields were comparable to those obtained with phenyl at R^1^, irrespective of the R^2^ functional group (Table 3, Entries 20–38). However, when *p*-toluidine or nitroaniline was used for R^3^, the yields were generally lower (Table 3, Entries 39–57). Notably, the substitution with diethylamine yielded relatively lower results in most cases (Table 3, Entries 5, 15, 34, 49).

## 3. Materials and Methods

All the chemicals were reagent grade and used as purchased. The Merrifield resin (loading capacity 1.29 mmol/g, 100-200 mesh) was phurchased from BeadTech (Seoul, Korea). The reactions were monitored by TLC analysis using Merck silica gel 60 F-254 thin layer plates (Merck, Darmstadt, Germany). Flash column chromatography was carried out on Merck silica gel 60 (230–400 mesh). The crude products, which were derived from the solid support, were purified by parallel chromatography using CombiFlash (Isco, Lincoln, NE, USA). The ^1^H NMR and ^13^C NMR spectra were recorded in d units relative to the deuterated solvent (CDCl_3_, DMSO-*d_6_*, etc.) as an internal reference by the Bruker 500 MHz NMR instrument (Bruker, Billerca, MA, USA). High-performance liquid chromatography (HPLC) system, specifically the Ultimate 3000, coupled with the Q-Exactive Focus quadrupole-Orbitrap MS (Thermo Fisher Scientific, Mass Spectrometry Based Convergence Research Institute, Kyungpook National University, Bremen, Germany). Mass Spectrometry Based Convergence Research Institute, Kyungpook National University The solid-phase synthesis was monitored by FT-IR using JASCO FT-IR 4600.

### 3.1. Synthesis of 4-Amino-2-(methylthio)-N-phenylthiazole-5-carboxamide (***7a***)

To a solution of **6a** (73.0 mg, 0.38 mmol) in H_2_O (0.5 mL) was added a solution of **5** (50 mg, 0.25 mmol) dissolved in acetone (2.5 mL) dropwise at room temperature. After the addition was completed, the reaction mixture was stirred at room temperature for 1 h. The mixture was added to LiOH (6.10 mg, 0.25 mmol) at room temperature and the reaction mixture was heated under reflux at 60 °C for 2 h. After cooling, methyliodide (15.85 µL, 0.25 mmol) in acetone was added dropwise. The mixture was stirred for 1 h at room temperature. The crude product was recrystallized from cold H_2_O to give compound **7a** (98%). As a solid, ^1^H NMR (500 MHz, CDCl_3_) δ 7.53–7.47 (m, 2H), 7.38–7.31 (m, 2H), 7.15–7.07 (m, 1H), 6.78 (s, 1H), 6.12 (s, 2H), 2.67 (s, 3H); ^13^C NMR (126 MHz, CDCl_3_) δ 169.67, 162.26, 137.99, 129.08, 124.40, 120.61, 93.61, 16.19. HRMS (ESI) *m/z* [M + H]^+^Calcd for C_11_H_12_N_3_OS_2_^+^ 266.0416; found 266.0418 (see Appendix A).

### 3.2. Synthesis of 2-(Methylthio)-6-phenylthiazolo [4,5-d]pyrimidin-7(6H)-one (***8aa***)

To a solution of **7a** (50 mg, 0.17 mmol) in EtOH (1 mL) was added triethylorthoformate (264.0 µL, 3.39 mmol) and CSA (7.9 mg, 0.03 mmol). The mixture was stirred at 60 °C for 2 h. After completion of the reaction (monitored by TLC), the reaction mixture was cooled down and then diluted with CH_2_Cl_2_, washed with brine, and dried over MgSO_4_. The solvent was removed, and the residue was purified by flash silica gel column chromatography (hexane/EtOAc, 1:1) to give **8aa** (37.6 mg, 79%), ^1^H NMR (500 MHz, CDCl_3_) δ 8.24 (s, 1H), 7.64–7.51 (m, 3H), 7.47–7.40 (m, 2H), 2.86 (s, 3H); ^13^C NMR (126 MHz, CDCl_3_) δ 177.52, 165.85, 156.12, 149.27, 136.60, 129.76, 129.62, 127.05, 117.66, 16.36. HRMS(ESI) *m/z* [M + H]^+^Calcd for C_12_H_10_N_3_OS_2_^+^ 276.0260; found 276.0281 (see Appendix A).

### 3.3. Synthesis of 2-(Methylsulfonyl)-6-phenylthiazolo [4,5-d]pyrimidin-7(6H)-one (***9aa***)

To a solution of 2-(methylthio)-6-phenylthiazolo [4,5-*d*]pyrimidin-7(6*H*)-one **8aa** (100 mg, 0.26 mmol) in CH_2_Cl_2_ was added *m*CPBA (132 mg, 0.57 mmol) at room temperature overnight. After completion of the reaction (monitored by TLC), the crude was quenched with NaHCO_3_ solution and extracted with CH_2_Cl_2_. The combined organic layer was dried over MgSO_4_. The solvent was removed, and the residue was purified by flash silica gel column chromatography (hexane:EtOAc:CH_2_Cl_2_) to give **9aa** (91 mg, 85%) as a solid; ^1^H NMR (500 MHz, CDCl_3_) δ 8.40 (s, 1H), 7.65–7.58 (m, 3H), 7.46 (dd, *J* = 8.1, 1.5 Hz, 2H), 3.50 (s, 3H); ^13^C NMR (126 MHz, CDCl_3_) δ 173.63, 165.55, 155.69, 150.03, 136.60, 129.76, 129.70, 127.52, 117.66, 42.07. HRMS (ESI) *m/z* [M + H] + Calcd for C_12_H_10_N_3_O_3_S_2_^+^ 308.0158; found 308.0160 (see Appendix A).

### 3.4. Synthesis of 2-(Butylamino)-6-phenylthiazolo [4,5-d]pyrimidin-7(6H)-One (***1aaa***)

To a solution of 2-(methylsulfonyl)-6-phenylthiazolo [4,5-*d*]pyrimidin-7(6*H*)-one **9aa** (100 mg, 0.32 mmol) in CH_2_Cl_2_ was added butylamine (0.072 mL, 0.72 mmol) and Et_3_N (0.1 mL, 0.72 mmol) at room temperature for 5 h. After completion of the reaction (monitored by TLC), the reaction mixture was extracted with CH_2_Cl_2_. The combined organic layer was dried over MgSO_4_. The solvent was removed and the residue was purified by flash silica gel column chromatography (hexane:EtOAc:CH_2_Cl_2_) to give **1aaa** (58 mg, 60%) as a yellow solid HRMS (ESI) *m/z* [M + H]^+^Calcd for C_15_H_17_N_4_OS_+_ 301.1118; found 301.1117 (see Appendix A).

### 3.5. Preparation of 4-Amino-N-(substituted)thiazole-5-carboxamide resin (***12a***) (R_1_ = Ph)

To a solution of **6a** (5.6 g, 29 mmol) in H_2_O (20 mL) was added a solution of **5** (2.9 g, 17.4 mmol) dissolved in acetone (100 mL) dropwise at room temperature. After the addition was completed, the reaction mixture was stirred at room temperature for 1 h. The mixture was added to LiOH (556 mg, 23.2 mmol) at room temperature and the reaction mixture was heated at reflux for 1 h. After cooling, the reaction solvent evaporated and was concentrated under reduced pressure and dried in a vacuum oven to give **7-Int II**. Merrifield resin **10** (4.5 g, 5.8 mmol, 1.29 mmol/g) was treated with crude **7-Int II** in acetone (100 mL) at room temperature. The reaction mixture was shaken for 10 h at room temperature and then filtered, washed several times with H_2_O, DMF, MeOH, and CH_2_Cl_2_, and dried in a vacuum oven to give 4-amino-*N*-(substituted)thiazole-5-carboxamide resin **12a** (6.3 g, 5.8 mmol) (see Appendix A).

### 3.6. Prepared of 2-(Methylthio)-6-phenylthiazolo [4,5-d]pyrimidin-7(6H)-one resin (***13aa***)

A mixture of resin **12a** (3.1 g, theoretically 2.85 mmol) and triethylorthoformate (4.7 mL, 28.5 mmol) in DMF:EtOH (2:1, 40 mL) was added to CSA (359 mg, 1.55 mmol) and shaken at 70 °C for 2 h, and then filtered, washed several times with H_2_O, DMF, MeOH, and CH_2_Cl_2_, and dried in a vacuum oven to give 2-(methylthio)-6-phenylthiazolo [4,5-*d*] pyrimidin-7(6*H*)-one resin **13aa** as a yellow resin (see Appendix A).

### 3.7. Prepared of 2-(Methylsulfonyl)-6-phenylthiazolo[4,5-d]pyrimidin-7(6H)-one resin (***14aa***)

To a mixture of 2-(methylthio)-6-phenylthiazolo[4,5-*d*]pyrimidin-7(6*H*)-one resin **13aa** (3.1 g, 2.85 mmol) in CH_2_Cl_2_ (100 mL) was added *m*CPBA (2.6 g, 11.4 mmol) at room temperature. The reaction resin mixture was shaken overnight and then filtered, washed several times with H_2_O, DMF, MeOH, and CH_2_Cl_2_, and dried in a vacuum oven to give **14aa** (3.08 g) as a yellow resin (see Appendix A).

### 3.8. Synthesis of 2-(Butylamino)-6-phenylthiazolo[4,5-d]pyrimidin-7(6H)-one (***1aaa***)

To a mixture of 2-(methylsulfonyl)-6-phenylthiazolo[4,5-*d*]pyrimidin-7(6*H*)-one resin **14aa** (0.33 g, 0.31 mmol) in CH_2_Cl_2_ (10 mL) was added butylamine (0.093 mL, 1.5 mmol) and Et_3_N (0.13 mL, 1.5 mmol) at room temperature. The reaction mixture was shaken for overnight and then filtered, and it was washed several times with MeOH and CH_2_Cl_2_. The organic solvent was removed, and the residue was purified by flash silica gel column chromatography (hexane:EtOAc) to give **1aaa** (75 mg, 81%) as a yellow resin (see Appendix A).

## 4. Conclusions

In summary, thiazolo-pyrimidinone derivatives have demonstrated a range of biologically relevant activities in previous studies. To further explore the structure–activity relationship (SAR) of thiazolo-pyrimidinones, we have developed a synthetic route for derivatives containing various functional groups, achieving yields ranging from 18% to 93% over four steps using solid-phase synthesis. Additionally, future research will focus on synthesizing derivatives of thiazolo-aminoamide **12** through solid-phase methods. We believe that the thiazolo-pyrimidinone derivative library will serve as a valuable resource for identifying potential hit compounds.

## Data Availability

The Appendix A is available free of charge on the MDPI Publication website: ^1^H NMR, ^13^C NMR and synthetic procedure of all compounds.

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
