# Peer review of "The Facile Solid-Phase Synthesis of Thiazolo-Pyrimidinone Derivatives"

_molecules, 2025, doi:10.3390/molecules30020430_

Round 1

Reviewer 1 Report

Comments and Suggestions for Authors

Dear Sir,

The article with the title of “The facile Solid-Phase synthesis of Thiazolo-pyrimidinone derivatives”is suggested to be published after some minor reivsions.

Minor revision as following: 

1.     Figure 1 middle structure should be thiazole derivative instead of thiofene derivatives.

In the Scheme 2. Solid-phase synthesis of thiazolopyrimidinone derivatives 1. The compound 11 should be the compound 7, which have been shown in Scheme 1 already. 

Author Response

The article with the title of “The facile Solid-Phase synthesis of Thiazolo-pyrimidinone derivatives”is suggested to be published after some minor reivsions.

Minor revision as following: 

(1).     Figure 1 middle structure should be thiazole derivative instead of thiofene derivatives.

Answer : We are conducting research with the goal of building various five-membered heterocyclic derivatives libraries through solid-phase synthesis. Figure 1 shows the results of the library we have constructed using solid-phase synthesis. In the manuscript, the following is newly mentioned: 'In our previous work, we have successfully constructed libraries of five-membered heterocyclic derivatives, including 4-substituted xanthine derivatives, 3-substituted thiazole derivatives, and 3-substituted thiophene derivatives, using solid-phase synthesis' (lines 55-58).

(2). In the Scheme 2. Solid-phase synthesis of thiazolopyrimidinone derivatives 1. The compound 11 should be the compound 7, which have been shown in Scheme 1 already. 

 Answer : The compound 11 presented in Scheme 2 has been modified to 7-Int II, and the original compound 11 is an intermediate that has not undergone methylation after the Thorpe-Ziegler reaction. Solid-phase synthesis proceeds by coupling 7-Int II with Merrifield resin 10. The synthesis process has been newly illustrated in Scheme 1.

Other changes:

  1. The author's name “Shanghui Hua” has been corrected to “Shuanghui Hua”

We made all revisions and corrections as far as we could.

Thank you for your consideration and kindness.

Sincerely yours,

Taeho Lee, Assistant Professor

College of Pharmacy, Research Institute of Pharmaceutical Sciences

Kyungpook National University

80 Daehak-ro, Buk-gu

Daegu 702-701, Korea

Tel) 82-53-950-8573; Fax) 82-53-950-8557; E-mail) tlee@knu.ac.kr

Reviewer 2 Report

Comments and Suggestions for Authors

See attached file

Comments on the Quality of English Language

Improvement in English language and some typing is needed.

Author Response

The authors describe the solid-phase synthesis of thiazolo-pyridinone derivatives, where they initially investigated an optimization protocol (in solution), which was then applied to Merrifield’s resin to obtain a series of the targeted compounds. Overall, the work is of interest, however, the presentation is poor, and several questions arise when reading the ms and the experimental part. My recommendation is that the authors should make some effort to improve the description part, and they should also revise some of the experimental descriptions given. Some detailed comments to better explain the recommendations given are listed/described below:

  1. Review again the introduction, especially as it concerns the previously synthesized scaffold, for example the following parts are not clearly presented:

1.1 4c reference is not complete.

Answer : The structure of the compound from reference 4c has been detailed further by adding “4” to Figure 1, and the IC50 data of the compound has been included in the Introduction

1.2 In line 48, thiazole core 1 is not correct. Compound 1 does not represent thiazole core. Please remove the number.

Answer : The '1' in that line 48 has been removed. Additionally, “the bioactivity of the thiazole core 1” has been revised to “the bioactivity of the thiazole fused heterocyclic compounds 2, 4”.

1.3 In Figure 1 there are three structures related to publications 5a, 5b, 5c (note that, contrary to what is described in line 46, there are no 5d, 5e references). Since these structures have been studied/reported before, it is better to clarify this issue in the legend of Figure 1, in order to be easily identified by the reader, and/or to better describe it in the text. Furthermore, in Figure 1 (line 42) you refer to 4c and then in Figure 2 (line 46), although you refer to the general forms (2, 3, 4), the given references (5a-c) refer to the structures presented in Figure 1. This is all somehow confusing, and one has to search the given literature (which also needs to be reviewed) in order to clarify the situation.

Answer : The reference range '5a-5e' has been corrected to '5a-5c.' The structures of references 5a, 5b, and 5c have been drawn in more detail in Figure 1. Additionally, the order of Figure 1 and Figure 2 has been switched.

1.4In Figure 2 replace “derivative” by “derivatives”

 Answer : “Derivative” has been changed to ”derivatives”

  1. In line 54 it is better to use the title “Results and Discussion”.

Answer : “Results” has been changed to ”Results and Discussion”

  1. Regarding the Thorpe-Ziegler cyclization step, you first synthesize 7-int, which was not isolated, thus directly applying the Thorpe-Ziegler cyclization process to this molecule, finally adding MeI to methylate the previously generated thiol group (thus producing 7). Although you did not isolate 7-int, it is not clear, from a presentation point of view, why you did not introduce this product in the synthetic scheme, as an intermediate (e.g. as the 7-int product of 5+6, in brackets). In any case, you should clarify, in the description part, that 7-int represents the non-isolated intermediate of the f irst reaction step (5+6 in acetone:H2O 5:1). You should also show that the reaction of 5+6 to give 7 represents three different steps, for example by adding numbers: e.g. 1. Acetone:H2O 5:1 (design 7-int in brackets); 2. LiOH; 3. MeI. As it is presented now, it is as if you added all the reagents together, in a one-pot reaction, which is not accurate.

Answer : The process has been categorized into three steps in Scheme 1. “5” and “6a” are mixed to generate “7-Int I” which is then used in the Torpe-Ziegler reaction to synthesize “7-Int II” This process has been depicted in Scheme 1. Additionally, the methylation of “7-Int II” using MeI to synthesize “7a” has been revised.

 “6” has been changed to ”6a

7” has been changed to ”7a

7-Int” has been changed to ”7-Int I

7-Int II” has been added

  1. It is preferable to change paragraph in line 65-66 (and not in line 74), because this is where you start describing the cyclization process.

Answer : The paragraph has been moved to line 91, below Table 1

  1. In line 66, position 5 refers to the position of the R2 substituent in the heterocycle 1. This is not compound 5, and thus it is suggested that you do not use bold typing.

Answer :

"Next, we explored various synthesis conditions to introduce different functional groups at position 5”

has been revised to

“Next, we optimized the solution-phase synthesis condition for an efficient solid-phase synthesis method. Through various cyclization reaction conditions, the process of introducing diversity factors to the R2 position was optimized."

  1. I don’t see the reason to name a compound 1aaa. You should use a natural number (see also the next comments/suggestions).

Answer : The alphabets of the diversity elements have been inserted into each step of Scheme 1

6” has been changed to ”6a

7” has been changed to ”7a

8” has been changed to “8aa

9” has been changed to ”9aa

  1. In the legend of Table 2, remove the room temperature conditions (c) (otherwise add these entries), and correct “catalyst”. The temperature in entry 3 was 80 oC ? Please review.

Answer :

The footnote “c Room temperature” has been removed.

The “80” in Entry 3 has been changed to “Reflux”

“catayst” has been changed to “catalyst”

  1. In Table 2, what is the reaction time that is required for the reactions to be completed (in the specified conditions), or the reaction times tested? This is not reported.

Answer : The reaction time for the cyclization has been added to Table 2.

  1. In the Materials and Methods section, for the conversion of 7 to 8 (lines 156-163), you report room temperature conditions in DMF and CSA as catalyst. This combination is not described in the main text, nor has it been added in Table 2. More importantly, this reaction was supposed to require relatively high temperatures and the best reaction conditions were supposed to be the use of EtOH and CSA and the reaction temperature at 60 oC (in 79% yield, as reported in Table 2). So, it is not clear why you describe the efficient synthesis in DMF/CSA at room temperature (in the materials and methods section).

 Answer : The solvent 'DMF' in the SI has been changed to 'EtOH.' Additionally, the following text has been revised:

“To a solution of 7 (50 mg, 0.17 mmol) in DMF (1 mL) was added triethylorthoformate (264.0 µL, 3.39 mmol) and CSA (7.9 mg, 0.03 mmol), the mixture was stirred at room temperature for 2 h, and then diluted with CH2Cl2, washed with brine, dried over MgSO4. The solvent was removed, and the residue was purified by flash silica gel column chromatography (hexane/EtOAc, 1:1) to give 8 (37.6 mg, 73%), HRMS(ESI) m/z [M+H]+Calcd for C12H10N3OS2+ 276.0260; Found 276.0281”

has been changed to:

“To a solution of 7a (50 mg, 0.17 mmol) in EtOH (1 mL) was added triethylorthoformate (264.0 µL, 3.39 mmol) and CSA (7.9 mg, 0.03 mmol), the mixture was stirred at 60 °C for 2 h. After completion of the reaction (monitored by TLC), the reaction mixture was cooled down and then diluted with CH2Cl2, washed with brine, dried over MgSO4. The solvent was removed, and the residue was purified by flash silica gel column chromatography (hexane/EtOAc, 1:1) to give 8aa (37.6 mg, 79%), 1H NMR (500 MHz, CDCl3) δ 8.24 (s, 1H), 7.64 – 7.51 (m, 3H), 7.47 – 7.40 (m, 2H), 2.86 (s, 3H). 13C NMR (126 MHz, CDCl3) δ 177.52, 165.85, 156.12, 149.27, 136.60, 129.76, 129.62, 127.05, 117.66, 16.36. HRMS(ESI) m/z [M+H]+ Calcd for C12H10N3OS2+ 276.0260; Found 276.0281”

  1. It is preferable to change paragraph in line 86-87 (instead of line 93) where you start describing the solid-phase synthesis part.

Answer : The paragraph

“Next, solid-phase synthesis was carried out based on the optimized solution-phase conditions. Merrifield resin and the synthesized intermediate 11 were stirred in acetone to form thiazolamide resin 12

has been revised to:

“Next, solid-phase synthesis was carried out based on the optimized solution-phase conditions. Merrifield resin and the synthesized intermediate 7-Int II were stirred in acetone to form thiazolo-aminoamide resin 12a

This has been placed below Table 2."

  1. In Scheme 2 correct “resin”. In Figure 3 correct “synthesis”

Answer : “rein” has been changed to ”resin

syntehsis” has been changed to ”synthesis

  1. Regarding the attachment of compound 11 to resin 10 (lines 182-192), what was the solvent that you finally used to attach 11 on the resin? It seems that you dissolved the residue (11) in acetone, however water should be present, as it cannot be evaporated (based on the procedure presented in the materials and methods part, lines 182-192). Could you clarify?

Answer : Water is used as the solvent for synthesizing “7-Int II”. However, since the use of water interferes with the reaction progress in solid-phase synthesis, after synthesizing “7-Int II” the solvent is removed, and the mixture is dried under vacuum for a certain period before reacting with Merrifield resin 10. This has been additionally noted in the Materials section.

  1. For the conversion of 12 to 13, in the Materials and Methods section you use only DMF (at 80 oC), instead of DMF-EtOH that you described in the text. Although this also seems to be an option, what did you end-up doing? In addition, these conditions (DMF/CSA/80oC) are not described in the discussion section (nor were they entered in Table 2). Please clarify this point, and correct accordingly (both in the Materials and Methods section and in the discussion part). In addition, the washings of the resin does not seem to be accurate. Did you finally wash the resin with DMF and then water? Why? In general, the Merrifilled resin, as all crosslinked polystyrene-type resins, is sequentially washed with solvents like DMF, DMF/H2O, DCM, and afterwards MeOH. Thus, some representative mixtures are: DMF/DCM/MeOH; DMF/DMF-H2O/DMF/DCM/MeOH; etc. I don’t see the reason to wash with DCM, MeOH and then DMF, and finally water. Maybe the resin was washed in the opposite direction (than the one reported) e.g. H2O/DMF/DCM/MeOH? (closer to the washings in line 205?). Please explain and/or correct accordingly (all washing procedures of the resins).

Answer : The error regarding the solvent and temperature in the Materials section has been corrected. As noted by the reviewer, both the DMF/CSA condition and the DMF:EtOH/CSA condition allow the reaction to proceed. However, the use of only DMF has the disadvantage of a longer reaction time, while the mixture of DMF and EtOH facilitates a faster reaction. By comparing the times in Table 2 for Entry 3 and Entry 4, it is clear that there is more than a fivefold difference in reaction time. We have revised the Materials section accordingly to reflect this process modification. Additionally, the error in the washing process of the resin has been corrected to the sequence of H2O-DMF-MeOH-CH2Cl2.

The revision is as follows:

“A mixture of resin 12 (3.1 g, theoretically 2.85 mmol), triethylorthoformate (4.7 mL, 28.5 mmol) in DMF (40 mL) and added CSA (359 mg, 1.55 mmol) was shaken at 80°C for 8 h, and then filtered, washed several times with CH2Cl2, MeOH, DMF, and H2O, and dried in a vacuum oven to give 2-(methylthio)-6-phenylthiazolo[4,5-d]pyrimidin-7(6H)-one resin 13 as a yellow solid” has been revised to:

“A mixture of resin 12a (3.1 g, theoretically 2.85 mmol), triethylorthoformate (4.7 mL, 28.5 mmol) in DMF:EtOH (2:1, 40 mL), and CSA (359 mg, 1.55 mmol) was shaken at 70 °C for 2 h, and then filtered, washed several times with H2O, DMF, MeOH, and CH2Cl2, and dried in a vacuum oven to give 2-(methylthio)-6-phenylthiazolo[4,5-d]pyrimidin-7(6H)-one resin 13aa as a yellow solid"

  1. In lines 142-144 the hplc equipment is described. Please specify in which part of your work this equipment was used (solution and/or solid-phase synthesis) and what was its exact use? It also appears that the completion of the solid-phase synthesis reactions was monitored only by FT-IR, possibly because of difficulties to cleave the attached molecules from resins 12-14 before the final step of nucleophilic attack. However, FT-IR is not of so high diagnostic value, especially since you follow the disappearance of the two small peaks for amine groups. Therefore, support of hplc in the characterization of the final crude products (before silica gel chromatography purification) would be useful to confirm the applicability of the suggested method. Could you explain the use of hplc and if it was also used to identify the purity of the crude products (before their silica gel purification)?

Answer : After the desulfonative nucleophilic substitution reaction in the solid-phase synthesis, all compounds were purified by silica gel chromatography. The HPLC system was used for HRMS analysis of the compounds. The HRMS data have been included in the SI.

  1. Were you able to identify/ensure the complete conversion of resin 13 to 14? Is it possible to get any information from the absorption bands of sulfones (in comparison with/verification of the absorption bands of the corresponding product synthesized in solution)?

Answer : the FT-IR shift of the sulfone group has been discussed in the manuscript, specifically in lines 123-125

“We oxidized sulfide using mCPBA to substitute nucleophile in the R3 position of thiazolo-pyrimidinone resin 13aa. In this process, new 1151 cm-1 and 1336 cm-1 sulfone stretches could be observed”

  1. For the reported yield of 81% in line 102, please specify that this is a four-step overall yield from resin 10 and reference to table 3. This is not easily recognized. Also, consider to rephase the phrase “perfect yield” or just remove “perfect”.

Answer : The figure in Table 3 has been newly revised, starting from Merrifield resin 10. Additionally, the footnote has been updated to include: 'b Four-step overall isolated yield from Merrifield resin 10 (loading capacity = 1.29 mmol/g).' To aid readers' understanding, the building blocks in Scheme 2 have been added with new numbering.

The word “perfect” has been removed.

  1. The structure 1 on line 131, just before Table 3, is different than the structure 1 on line 50, in the sense that the structure on line 131 is closer to the synthesized compounds. However, you cannot have two different structures with the same number, and also two different types of R1 substituents (those in line 50, and those in line 131). Besides, the structure on line 131 is more representative of the targeted molecules (as both the solution optimization process and the solid-phase synthesis were evaluated with Ph-type R1 substituents). I would recommend changing structure 1 on line 50 with the structure of line 131, or, if you would like to present also the more general form, draw both structures in Figure 1 (with an arrow between them) and number as 1 only the second one (the one that is currently presented on line 131).

Answer : To enhance readers' understanding, the figures in Scheme 1, Scheme 2, and Table 3 have been unified under the original compound 1(line 50). Additionally, the functional groups R1, R2, and R3 have been placed under their respective numbers in the schemes for better clarity. Furthermore, the alphabet labels for the R2 functional group in the solid-phase synthesis have been added to Scheme 2.

1aaa (R1 = Ph, R2 = H, R3 = NHBu) 81%

1aah (R1 = Ph, R2 = H, R3 = SPr)77%

1aba (R1 = Ph, R2 = Me, R3 =NHBu)85%

  1. The authors could also consider drawing Scheme 4 as a general strategy for the solid-phase synthesis of type 1 molecules, showing the R1, R2, R3 groups, resulting in the general form for the targeted compounds (1), referring to Table 3 for the exact type of R1, R2, R3 substituents. In this case, they could describe the conversion of 12 to 13 for example using “triethyl orthoformate or triethyl orthoacetate” (or the corresponding condensed formulas and the solvents/ratio) and for the conversion of 14 to 1 they could use letters a-j (to describe the nucleophiles used).

Answer : Instead of drawing Scheme 4, the functional groups have been added below the compound numbering in Scheme 2 for better clarity. Corresponding details have also been added in lines 125-129.

The content is as follows:
'We oxidized the sulfide using mCPBA to substitute the nucleophile in the R3 position of thiazolo-pyrimidinone resin 13aa. In this process, new sulfone stretches at 1151 cm-1 and 1336 cm-1 were observed. Then, 1aaa was synthesized in an 81% yield through a nucleophilic substitution reaction using butylamine as the nucleophile and triethylamine as the base. Additionally, thiol-type nucleophiles at R3 could be synthesized with a 77% yield. To introduce the methyl group at R2, 13ab was synthesized using triethyl orthoacetate, and butylamine was substituted with nucleophiles after sulfide oxidation to synthesize 1aba with an 85% yield.

  1. What was the procedure that was followed for the reaction of h-j nucleophiles with resin 14? Did you use NEt3 as a base? The procedure is not described in materials and methods section.

Answer : All nucleophiles for R3 were reacted under the same conditions as described in the materials section. This information has also been added to Scheme 2

  1. The nomenclature of some molecules is written in different ways. E.g. triazolo pyrimidinone or triazolopyrimidinone; thiazolo-amino-amide or thiazole amino amide or thiazoloaminoamide. It is suggested that one way of writing be used.

Answer : “thiazolopyrimidinone” was changed to “thiazolo-pyrimidinone”

        “thiazoloaminoamide” was changed to "thiazolo-aminoaminde”

Other changes:

  1. The author's name “Shanghui Hua” has been corrected to “Shuanghui Hua”

We made all revisions and corrections as far as we could.

Thank you for your consideration and kindness.

Sincerely yours,

Taeho Lee, Assistant Professor

College of Pharmacy, Research Institute of Pharmaceutical Sciences

Kyungpook National University

80 Daehak-ro, Buk-gu

Daegu 702-701, Korea

Tel) 82-53-950-8573; Fax) 82-53-950-8557; E-mail) tlee@knu.ac.kr

Reviewer 3 Report

Comments and Suggestions for Authors

In the article titled: “The facile Solid-Phase synthesis of Thiazolo-pyrimidinone derivatives “, the authors discussed the synthesis in solution and in solid phase of thiazole, a five-member ring, fused with a pyrimidinone, a six-member ring. In the introduction the authors underlined the importance of heterocycles in the development of new drugs. The introduction, however, is too poor and requires greater depth.

There are several works in the literature on the synthesis of 5-membered heterocycles that the authors have not mentioned and discussed, please introduce and discuss them in the introduction: 10.1002/chem.201402519, 0.1002/ejoc.201100346, 10.1021/acs.joc.8b03055, 10.1039/c2ob07172j. Discuss the synthesis procedures and what is their disadvantage and compare them with the synthesis they propose.

In the results the authors have reported the image of scheme 1 but it is not commented on. Please introduce this when discussing the results.

FT-IR spectrum, which solvent did you use to prepare the samples?

Please compare the FT-IR data with H and C-NMR spectrum and discuss them in the manuscript.

Discussion of the results is poor. Please, add a paragraph for the discussion of the results and discuss them in depth.

Minor revisions

Page 2, line 60, replace “7” with “7

Page 6, line 86, the authors wrote “the final compound 1aaa with a 70% yield (Scheme 3)”, I think the scheme 3 is missing. Could you introduce it, please?

Figure 4 and table 3 have moves before the conclusions.

Author Response

Reviewer 3’s Comments:

(1) In the article titled: “The facile Solid-Phase synthesis of Thiazolo-pyrimidinone derivatives “, the authors discussed the synthesis in solution and in solid phase of thiazole, a five-member ring, fused with a pyrimidinone, a six-member ring. In the introduction the authors underlined the importance of heterocycles in the development of new drugs. The introduction, however, is too poor and requires greater depth.

Answer : Examples of five-membered heterocyclic compounds with biological activities have been newly added to the introduction(line 24-28).

“Many pentagonal heterocyclic compounds are being used as key structures to identify hit compounds, including thiazole derivatives with α-amylase inhibitory activity2g, thiophene derivatives with potential anticancer efficacy2h, imidazole derivatives with potential antimalarial activity based on SAR studies2i, and oxazole derivatives with potential hypoglycemic effects2j”

(2) There are several works in the literature on the synthesis of 5-membered heterocycles that the authors have not mentioned and discussed, please introduce and discuss them in the introduction: 10.1002/chem.201402519, 0.1002/ejoc.201100346, 10.1021/acs.joc.8b03055, 10.1039/c2ob07172j. Discuss the synthesis procedures and what is their disadvantage and compare them with the synthesis they propose.

Answer : Several additional methods using solid-phase synthesis have been newly included in the introduction. Additionally, the paper you recommended on the synthesis of peptidomimetics has been added to the references(line 36-40)

“Until now, various synthesis methods such as heterocyclic compounds and peptides using solid phase synthesis have been introduced. Solid phase synthesis methods are used in various synthesis fields such as macrocycle synthesis3b, peptidomimetics synthesis with chirality3c-3e, and pyrimidine derivatives synthesis using microwaves3f and are worth utilizing”.;

however, I was unable to locate the article 0.1002/ejoc.201100346

(3) In the results the authors have reported the image of scheme 1 but it is not commented on. Please introduce this when discussing the results.

Answer : Additional information related to Scheme 1 has been included in the Results and Discussion section (lines 73, 108).

(4) FT-IR spectrum, which solvent did you use to prepare the samples?

Answer : The FT-IR measurements were performed using an ATR instrument. Solvent was not used; instead, the resin was washed, vacuum-dried, and then measured in the solid state

(5) Please compare the FT-IR data with H and C-NMR spectrum and discuss them in the manuscript. Discussion of the results is poor. Please, add a paragraph for the discussion of the results and discuss them in depth.

 Answer : The changes in the IR spectra (amine, amide, sulfone) of compounds 12a, 13aa, and 14aa have been added to the discussion. Additionally, the introduction of a methyl group at R2 and the thiol substitution reaction at R3 related to solid-phase synthesis have been included in Scheme 2, with the corresponding details added to lines 122-129. Additionally, for FT-IR and NMR, the amine shift at 6.12(s, 2H) in the NMR of “7a” and the FT-IR peaks at 3482 and 3351  cm-1 in “12a”are identical. When “7a” is reacted to synthesize “8aa”, the 6.12(s, 2H) shift disappears in the NMR. Similarly, when synthesizing “13aa” from “12a”, the 3482 and 3351 cm-1 peaks disappear. However, due to the extensive nature of this analytical data, it has not been included in the manuscript. Instead, this information can be found in the SI.

We kindly ask for your understanding

 “We oxidized sulfide using mCPBA to substitute nucleophile in the R3 position of thiazolo-pyrimidinone resin 13aa. In this process, new 1151 cm-1 and 1336 cm-1 sulfone stretches could be observed, and then 1aaa could be synthesized in an 81% yield through the nucleophilic substitution reaction using butylamine as a nucleophile and triethylamine as a base. In addition, thiol types of nucleophiles of R3 could be synthesized at a yield of 77%. To introduce the methyl group of R2, 13ab was synthesized using triethylorthoacetate, and then butylamine was substituted with nucleophiles after oxidation of sulfide to synthesize 1aba with an 85% yield”

Minor revisions

Page 2, line 60, replace “7” with “7” : “7” was changed to “7a

Page 6, line 86, the authors wrote “the final compound 1aaa with a 70% yield (Scheme 3)”, I think the scheme 3 is missing. Could you introduce it, please? : “(Scheme 3)” was changed to '(Scheme 1)' in line 86.

Figure 4 and table 3 have moves before the conclusions. : “Figure 4” and “Table 3” was moved before the conclusions.

Other changes:

  1. The author's name “Shanghui Hua” has been corrected to “Shuanghui Hua”

We made all revisions and corrections as far as we could.

Thank you for your consideration and kindness.

Sincerely yours,

Taeho Lee, Assistant Professor

College of Pharmacy, Research Institute of Pharmaceutical Sciences

Kyungpook National University

80 Daehak-ro, Buk-gu

Daegu 702-701, Korea

Tel) 82-53-950-8573; Fax) 82-53-950-8557; E-mail) tlee@knu.ac.kr

Round 2

Reviewer 2 Report

Comments and Suggestions for Authors

Signficant improvements have been made. Please consider the following minor points.

Reference 4c is not yet complete. The authors names are not included “+++ Med. 339 Chem. Commun., 2017, 8, 1655–1658”. Review all references in accordance with the journal guidelines.

Authors should carefully check the numbering of molecules. For example, with the corrections made, the numbering of compounds now starts from compound 2.

In the Materials and Methods Section, you report the synthesis of resins 13aa and 14aa as yellow solids (lines 277, 285 of the revised version). But they are not solids. They are resins. 

Some typos still need to be corrected.

Author Response

January 16, 2025

Ms. Gracia Zhong, Assistant Editor

Molecules

Molecules Editorial Office

Klybeckstrasse 64, 4057 Basel

Switzerland

Re: The facile Solid-Phase synthesis of Thiazolo-pyrimidinone derivatives (molecules-3407108)

Dear Ms. Gracia Zhong:

Thank you very much for your e-mail informing us our manuscript is publishable in Molecules after major revision. I also want to express my thanks to reviewers for their careful review of our work.

What follows is my response to reviewer’s critique with the explanation of the changes implemented in the paper and a rebuttal when appropriate.

Reviewer 2’s Comments:

Signficant improvements have been made. Please consider the following minor points.

Reference 4c is not yet complete. The authors names are not included “+++ Med. 339 Chem. Commun., 2017, 8, 1655–1658”. Review all references in accordance with the journal guidelines.

Answer : Reference format has been changed.

Authors should carefully check the numbering of molecules. For example, with the corrections made, the numbering of compounds now starts from compound 2.

Answer : For compound 2 in Figure 1, it has the same structure as the core compound of our study. However, since compound 2 is not one of our results, the numbering '1' was introduced for the core compound structures in Figure 2 to avoid confusion

In the Materials and Methods Section, you report the synthesis of resins 13aa and 14aa as yellow solids (lines 277, 285 of the revised version). But they are not solids. They are resins. 

 Answer : “Solid” has changed to “resin”

Some typos still need to be corrected.

 Answer : “shceme 1” has been changed to “scheme 1”

“R2” has been changed to “R2

Other changes:

  1. Reference format has been changed

We made all revisions and corrections as far as we could.

Thank you for your consideration and kindness.

Sincerely yours,

Taeho Lee, Assistant Professor

College of Pharmacy, Research Institute of Pharmaceutical Sciences

Kyungpook National University

80 Daehak-ro, Buk-gu

Daegu 702-701, Korea

Tel) 82-53-950-8573; Fax) 82-53-950-8557; E-mail) tlee@knu.ac.kr

Reviewer 3 Report

Comments and Suggestions for Authors

The authors improved the manuscript. There are some errors. However, I suggest the publication in this journal.

Page 3, line 79, replace "shceme 1" with "scheme 1";

Page 5, line 116, replace "Râ‚‚" with "Râ‚‚"

Author Response

January 16, 2025

Ms. Gracia Zhong, Assistant Editor

Molecules

Molecules Editorial Office

Klybeckstrasse 64, 4057 Basel

Switzerland

Re: The facile Solid-Phase synthesis of Thiazolo-pyrimidinone derivatives (molecules-3407108)

Dear Ms. Gracia Zhong:

Thank you very much for your e-mail informing us our manuscript is publishable in Molecules after major revision. I also want to express my thanks to reviewers for their careful review of our work.

What follows is my response to reviewer’s critique with the explanation of the changes implemented in the paper and a rebuttal when appropriate.

Reviewer 3’s Comments:

Page 3, line 79, replace "shceme 1" with "scheme 1";

Answer : “shceme 1” has been changed to “scheme 1”

Page 5, line 116, replace "Râ‚‚" with "Râ‚‚"

Answer : “R2” has been changed to “R2

Other changes:

  1. Reference format has been changed

We made all revisions and corrections as far as we could.

Thank you for your consideration and kindness.

Sincerely yours,

Taeho Lee, Assistant Professor

College of Pharmacy, Research Institute of Pharmaceutical Sciences

Kyungpook National University

80 Daehak-ro, Buk-gu

Daegu 702-701, Korea

Tel) 82-53-950-8573; Fax) 82-53-950-8557; E-mail) tlee@knu.ac.kr
